# Which Aspects of Psychological Resilience Moderate the Association between Deterioration in Sleep and Depression in Patients with Prostate Cancer?

**DOI:** 10.3390/ijerph19148505

**Published:** 2022-07-12

**Authors:** Christopher F. Sharpley, David R. H. Christie, Vicki Bitsika

**Affiliations:** 1Brain-Behaviour Research Group, School of Science & Technology, University of New England, Armidale, NSW 2351, Australia; david.christie@genesiscare.com (D.R.H.C.); vicki.bitsika@une.edu.au (V.B.); 2Genesiscare, John Flynn Private Hospital, Tugun, QLD 4224, Australia

**Keywords:** cancer, oncology, prostate, depression, sleep, resilience

## Abstract

This study aimed to investigate the moderating effect of psychological resilience on sleep-deterioration-related depression among patients with prostate cancer, in terms of the total score and individual symptoms. From a survey of 96 patients with prostate cancer, 55 who reported a deterioration in their sleep quality since diagnosis and treatment completed the Zung Self-Rating Depression Scale, Connor–Davidson Resilience Scale, and the Insomnia Severity Index. Moderation analysis was conducted for the scale total scores and for the ‘core’ symptoms of each scale within this sample, based on data analysis. Interaction analysis was used to identify key associations. The moderation analysis suggested that psychological resilience moderated the depressive effect of sleep deterioration that patients reported occurred after their diagnosis and treatment and did so at the total and ‘core’ symptom levels of being able to see the humorous side of things and to think clearly when under pressure, but there was an interaction between this moderating effect, the strength of psychological resilience, and severity of sleep deterioration. Although it appears to be a successful moderator of depression arising from sleep deterioration that was reported by patients with prostate cancer, the effectiveness of psychological resilience is conditional upon the severity of patients’ sleep difficulties and the strength of their psychological resilience. Implications for the application of resilience training and concomitant therapies for patients with prostate cancer with sleep difficulties and depression are discussed.

## 1. Introduction

Many patients with prostate cancer (PCa) also suffer from depression [1], with meta-analytic data suggesting that over 16% of these men suffer significant depressive symptoms [2]. This depression carries an added disease burden because PCa patients with depression are 6.5 times more likely to suicide than the general population during the first 6 months following initial PCa diagnosis [3], have an odds ratio of 4.45 for emergency room visits, 3.22 for hospitalisations, 1.71 for outpatient visits, and increased costs for health care, compared with other men with prostate cancer but who are not depressed [4]. The identification of the precursors to this depression, and also the possible ‘buffers’ against it, is a key aspect of clinically focussed research into depression among patients with PCa.

Receiving a diagnosis of cancer is stressful [5] and may affect physiological functioning in patients with PCa by hyper-arousal of the sympathetic nervous system [6]. One manifestation of that arousal may be changes in sleeping patterns, which are part of the diagnostic criteria for both generalised anxiety disorder and major depressive disorder [7]. There are data indicating that patients with PCa have an elevated prevalence of sleeping difficulties [8] and some suggest that this increase in sleeping difficulties may be associated with greater severity of depression in these men [9]. However, although sleeping difficulties have been found to correlate with depression in the general community [10], there is potential for a confounding factor due to the overlap between sleeping difficulties and the diagnostic criteria for Major Depressive Disorder, which include the symptom ‘insomnia or hypersomnia nearly every day’ [7]. Therefore, studies on the possible effect of sleeping difficulties on depression among patients with PCa need to consider how to reduce that potential confounding factor. 

In terms of those factors which may help patients with PCa avoid or reduce the severity of their depression, psychological resilience (PR) (which refers to an individual’s capacity to cope with stressors and to resist the harmful effects of future negative events [11]) has been shown to reduce the severity of depression induced by stressful events among older people [12] and has an inverse relationship with depression in patients with PCa [13]. As PR reduces autonomic responses to stressors [14], it helps the individual to return to calm and optimism when faced with stressors such as old age [15], chronic pain [16], terrorist attacks [17], and events such as the COVID-19 lockdowns [18]. Its association with the autonomic nervous system may also include alleviation of sleeping difficulties that arise from hyper-arousal of the sympathetic nervous system [19]. Clearly, PR has a potential role in the prevention and treatment of depression, particularly depression that is associated with sleeping problems [20,21]. 

Although there is some evidence (noted above) that sleep difficulties are related to depression among patients with PCa and that PR can have some protective effects on the development and severity of depression among these men (including depression arising from sleep difficulties), most previous studies have been conducted on total scores from self-report instruments or clinical interviews for depression, measures of sleep quality, or PR. However, these constructs are heterogeneous, and typical scales designed to measure them include a variety of symptoms and indicators. Examination of these constructs at the individual symptom/indicator level, as well as at the total scale score level, can provide novel and useful information for research and clinical purposes, where ‘individuation’ of a patient’s symptom profile for depression, sleep difficulties, and PR can provide a more effective set of therapy ‘targets’ [22,23]. Therefore, to further the effective clinical application of PR in patients with PCa who are depressed and whose mood state has been adversely affected by sleep difficulties, this study aimed to extend the current understanding of the association between sleep, PR, and depression in patients with PCa by investigating how PR might moderate the depressive effects of sleeping difficulties that have arisen in patients with PCa following their diagnosis and treatment for their illness. In addition to this novel approach to this issue, data were to be analysed at total scale score and individual symptom/indicator levels, which has hitherto not been undertaken when addressing these associations. 

## 2. Methods

### 2.1. Participants

From a previously recruited sample of 96 patients with PCa from treatment centres in South East Queensland, Australia, who were invited to participate in a study ‘about how you feel’ [24], 55 who reported a deterioration in their quality of sleep following diagnosis and treatment were selected for a study regarding the association between psychological resilience, deterioration in sleep, and depression.

All participants had biopsy-proven prostate cancer and were attending for either treatment or follow-up after previous treatment. Treatments included radiotherapy and/or surgery and hormone therapy when required. Other inclusion criteria were that (i) the diagnosis of prostate cancer was proven histologically, and (ii) all of the treatment options were properly considered by patients via discussion with their GP, a radiation oncologist, and a urologist. Unwillingness to participate in the study was the only exclusion criterion. The upper section of Table 1 describes the background data for this sample of 55 men.

### 2.2. Measures

Background questionnaire: age (in years), living situation (with wife/partner, widowed, separated/divorced, never married), month and year of the first diagnosis, past treatments, and current treatments (radiotherapy, surgery, hormone therapy, none), the present status of their cancer (cancer still present and undergoing initial treatment, no obvious sign of cancer (in remission), cancer reoccurring after previous treatment). 

Depression: The Zung Self-Rating Depression Scale (SDS) [25] measures 20 symptoms associated with major depressive disorder (MDD) [7]. Responses are made for ‘the last two weeks’ on a four-point scale, for ‘none or a little of the time’ (a score of 1), ‘some of the time’ (2), ‘good part of the time’ (3), and ‘all or almost all the time’ (4). Total scores are from 20 to 80, and SDS scores of 40 or above indicate the presence of ‘clinically significant depression’ [26] (p. 335). Split-half reliability for the SDS has been reported as 0.81 [25] 0.79 [27] and 0.94 [28]; internal consistency is 0.84 in PCa men [29]. The SDS has been shown to possess stronger validity than the Beck Depression Inventory and the MMPI Depression Scale in male psychiatric inpatients [30]. The SDS contains two items that relate to sleep and fatigue issues—namely, Item 4: ‘*I have trouble sleeping at night’* and Item 10: ‘*I get tired for no reason’*. Inclusion of these in any test of the association between the sleeping difficulties and the SDS total score might confound that relationship, and so the SDS total score was recalculated to exclude these two items to produce a measure of SDS minus sleep, called the ‘SDS-Sleep’ score.

The Insomnia Severity Index (ISI) [31] measures seven aspects of sleep functioning—namely, the difficulty of sleep onset, sleep maintenance, early morning awakening problems, sleep dissatisfaction, interference of sleep difficulties with daytime functioning, noticeability of sleep problems by others, and distress caused by the sleeping difficulties. Participants respond to the ISI on a 5-point Likert scale for each item, where ‘no problems’ is a score of 0, up to ‘a very severe problem’ (=4), giving a total score from 0 to 28. Internal consistency for the ISI is 0.74 [31], and this scale has shown good validity in a sample of patients with prostate cancer [32].

The Connor–Davidson Resilience Scale (CDRISC) [33] consists of 25 items such as ‘I like a challenge’, ‘When things look hopeless I don’t give up’, ‘I bounce back after illness or hardship’, and ‘I am able to adapt to change’ [33]. Responses are given on the 5-point scale of ‘not true at all’ (0), ‘rarely true’ (1), ‘sometimes true’ (2), ‘often true’ (3), and ‘true nearly all of the time’ (4) for how the respondent felt over the past month. This produces a total score between 0 and 100. Higher scores indicate greater resilience. Scores on the CDRISC are significantly correlated (0.83) with total scores on the Kobasa Hardiness Measure and negatively correlated with total scores on the Perceived Stress Scale (−0.76). Reliability is also sound (Cronbach alpha = 0.89) and test–retest reliability = 0.87 [33]; a previous study with PCa patients reported internal consistency of 0.922 [34]. 

### 2.3. Procedure

Patients received a Participant Information Statement, Background Questionnaire, SDS, CDRISC, and two copies of the ISI, and were asked to fill out the SDS and CDRISC, and one copy of the ISI for how they felt during the last two weeks, and the second copy for how they felt before receiving their diagnosis of prostate cancer. This procedure (known as the ‘retrospective pre-test’ procedure [35]) enables the change in sleep difficulties over time to be calculated in a different manner than is used in the traditional pre-test versus post-test design. The retrospective pre-test avoids such sources of invalidity as history, maturation, and testing artefacts, which can accompany the traditional pre- vs. post-test methodology [36], and has been used in previous studies of depression in patients with PCa [37]. Approval for this study was received from the UnitingCare Health Human Research Ethics Committee (Approval number 2013.32.104) in accordance with the Helsinki Declaration of 1964 and confirmed in 2013. Written informed consent was obtained from all participants. 

### 2.4. Statistical Analyses

Data were analysed via SPSS 25. Subtraction of the ISI score ‘before diagnosis’ from the ISI score of the ‘last two weeks’ produced a change in ISI score, with a positive score reflecting an increase in sleep problems, a negative score indicating a decrease in sleep problems, and a score of zero signifying no change in sleeping problems. As this sample was restricted to those patients with PCa who reported a deterioration in their sleep quality, all had positive scores for change in ISI. Frequencies described the sample’s background and PCa status. Data were checked for normality, and internal consistency (Cronbach alpha) was calculated for each scale. Pearson correlation coefficients were used to test the significant associations between the demographic variables and the depression, PR, and sleep variables, and also the relationships between the SDS, SDS-Sleep, CDRISC, and ISI change scores. Moderation analysis, which identifies ‘when, or under what circumstances’ [38] (p. 47) a variable exerts a moderating effect upon a dependent variable, was applied to describe the interaction between PR, SIS, and SDS-Sleep. Pearson correlations and hierarchical regression were used to identify the core components of the ISI and CDRISC, and item–total correlations identified which SDS-Sleep items were the core aspects of that scale. The moderating effect of PR upon sleep-related depression at this core level was determined via hierarchical regression and Hayes’ PROCESS analysis [39].

## 3. Results

### 3.1. Background Data

Internal consistency (Cronbach alpha) for the 55 PCa patients was as follows: SDS = 0.841, SDS-Sleep = 0.818, CDRISC = 0.923, ISI = 0.919 at the time of the survey and 0.914 for before diagnosis. None of the background variables (age, living situation, time since diagnosis, past and current treatments, current cancer status) showed any significant correlations with the SDS score, SDS-Sleep score, CDRISC score, either of the ISI scores, or the change in ISI scores (all *p* > 0.05). The SDS, SDS-Sleep, ISS, and CDRISC data did not require normalisation. The data from each scale appear in the lower section of Table 1. 

#### 3.1.1. Influence of PR on the Association between Sleep Difficulties and Depression: Total Scores

There was a significant correlation between increase in sleep problems in the ISI and SDS total score (*r* (55) = 0.370, *p* = 0.005), and also for the SDS-Sleep score (*r* = 0.303, *p* = 0.025), as well as between CDRISC score and SDS score (*r* = −0.543, *p* < 0.001) and SDS-Sleep score (*r* = −0.527, *p* < 0.001), but not between increase in ISI score and CDRISC score (*r* = 0.045, *p* = 0.738).

To test the hypothesis that SDS-Sleep in these patients with PCa was a function of changes in sleep quality but that this relationship was moderated by PR, a hierarchical regression analysis was conducted. In the first step, the ISI change score and CDRISC score were included and accounted for a significant amount of the variance in patients’ SDS-Sleep score: R^2^ = 0.422, *F*(2 50) = 18.244, *p* < 0.001. To avoid potentially problematic high multicollinearity with the interaction term, the variables were centred, and an interaction term between ISI change score and CDRISC score was created [40] and added to the regression model. This model accounted for a significant additional proportion of the variance in patients’ SDS-Sleep scores, ΔR^2^ = 0.212, Δ*F*(1, 49) = 28.345, *p* < 0.001, *b* = 2.311, *t*(49) = 5.324, *p* < 0.001, indicating that PR moderated the association between changes in the sleep quality of patients and their SDS-Sleep score.

The interaction plot (Figure 1) depicts the relative effects of PR on sleep-quality-induced SDS-Sleep (sleep quality deterioration is centred on the horizontal axis). High levels (i.e., 1 SD above the mean) of PR exerted a protective effect on the depressive effects of decreases in sleep quality, compared with lower levels of PR, across the range of deteriorations in sleep quality. However, that protection was most powerful when deterioration in sleep quality was minor, less when it was at the mean level, and least when sleep quality deterioration was at its highest level. This trend was consistent across all levels of PR, ranging from low (−1 SD below the mean CDRISC score), to mean CDRISC level, to high PR (1.0 SD above the mean CDRISC score).

#### 3.1.2. Influence of PR on the association between Sleep Difficulties and Depression: Specific Symptoms

##### Breakdown of Full-Scale Scores to Core Elements

However, these results are for the total scores on the ISI, CDRISC, and SDS-Sleep, whereas these are heterogeneous collections of various aspects of sleeping difficulties, PR, and depression, respectively. As mentioned in the Methods section, the ISI is comprised of seven items that measure different aspects of sleep quality. Pearson correlation analysis indicated that not all of the scores for changes in these seven items were significantly correlated with the SDS-Sleep score. At the Bonferroni-corrected *p* value of 0.05/7 = 0.007, only ISI change scores for items 1 (*Difficulty in falling asleep*: *r* = 0.343, *p* = 0.004), 3 (*Waking up too early*: *r* = 0.397, *p* = 0.003), and 5 (*My change in sleep difficulties is noticeable to others*: *r* = 0.378, *p* = 0.004) were statistically significantly associated with SDS-Sleep total score. When entered into a hierarchical regression in order of their correlation strength, only ISI change score for Item 1 made a significant contribution to the variance in the SDS-Sleep score (R squared = 0.118, *F* for change (1,53) = 7.084, *p* = 0.010), but neither ISI Item 3 nor Item 5 made a significant additional contribution to the variance in SDS-Sleep score.

Similarly, the CDRISC contains 25 items, each of which taps a different aspect of PR. Pearson correlations between SDS-Sleep and those components identified the four CDRISC items that were most powerfully associated with SDS-Sleep: Item 14 (*When I’m under pressure, I can focus and think clearly*: *r* = −0.624, *p* < 0.001); Item 6 (*I can see the humorous side of things*: *r* = −0.585, *p* < 0.001); Item 8 (*I tend to bounce back after illness or hardship*: *r* = −0.563, *p* < 0.001); and Item 22 (*I am in control of my life*: *r* = −0.545, *p* < 0.001). Hierarchical regression indicated that CDRISC Item 14 made the largest significant contribution to SDS-Sleep variance (R squared = 0.389, *F* for change (1,53) = 33.797, *p* < 0.001), and Item 6 added a further 0.077 to the R square (R squared = 0.466, *F* for change (1,52) = 7.482, *p* = 0.009). Neither CDRISC Item 8 nor Item 22 made any significant extra contribution to the variance in SDS-Sleep. A new variable composed of CDRISC items 6 and 14 was calculated and termed ‘CDRISC-Core’, which consisted of patients’ ability to see the humorous side of events, plus their ability to think clearly when under pressure. Finally, to identify the core elements of the SDS-Sleep scale, individual item–total correlations and the effect on the internal consistency of the total scale if an item was deleted were examined. Table 2 presents the results of that analysis and indicates that six of the SDS-Sleep items (shown in bold) may be considered to be the key aspects of depression (minus two sleep- and fatigue-related items), as reported by this sample of patients with PCa. These six items were then combined into a new variable that was named ‘SDS-Core’.

### 3.2. Associations between Core Elements of Sleep Change, PR, and Depression

There was a significant correlation between sleep change Item 1 in the ISI and the SDS-Core (*r* (55) = 0.284, *p* = 0.029), as well as between CDRISC-Core and SDS-Core (*r* = −0.512, *p* < 0.001), but not between sleep change Item 1 on the ISI and CDRISC-Core (*r* = −0.156, *p* = 0.240).

Moderation analysis was conducted for these core elements via hierarchical regression based on correlation coefficient strength. Sleep change Item 1 and CDRISC-Core accounted for a significant amount of the variance in patients’ SDS-Core: R^2^ = 0.305, *F*(2, 52) = 12.286, *p* < 0.001. As mentioned above, the variables were also centred, and the interaction term between sleep change Item 1 and SDS-Core was added to the regression model. As this model accounted for a significant additional proportion of the variance in patients’ SDS-Sleep scores (ΔR^2^ = 0.515, Δ*F*(1, 51) = 23.755, *p* < 0.001, *b* = 1.579, *t*(49) = 4.874, *p* < 0.001), it was concluded that the core aspects of the CDRISC moderated the association between patients’ sleep change Item 1 and their SDS-Core.

The interaction plot (Figure 2) reflected the findings shown in Figure 1; that is, high levels of CDRISC-Core might have protected patients from the effects of decreases in sleep quality Item 1 on SDS-Core and might have done so most powerfully for those patients who had CDRISC-Core scores 1.0 SD above the mean, compared with their peers whose CDRISC-Core scores were average or below-average level. Additionally, similar to Figure 1, the moderating effects of CDRISC-Core were strongest when the deterioration in sleep quality was less rather than more severe. 

## 4. Discussion

These results extend previous understanding of the association between sleep, PR, and depression in patients with PCa by noting that those men who reported a deterioration in sleep quality were also significantly more likely to become depressed, even when the two SDS items relating to sleep and fatigue were removed from that scale. Previous findings have indicated only tentative correlations between sleep problems and depression among patients with PCa [8,9], but these data provide a more detailed account of how that association might occur. Additionally, to extend the current understanding of this relationship, patients received some PR-based moderating effects on their levels of sleep-related depression severity, which had not been previously reported. The second major aim of this study was to extend those initial findings from the previous literature that focused solely on total scale scores by examining these relationships at the individual ‘core’ symptom level as well as the total scale level. Our results indicated that moderation of the depressive effects of decreased sleep quality from diagnosis/treatment to the time of the present survey was found at the total scale level and also at the ‘core’ element level for sleep quality, PR, and depression. The nuanced explanation of this association resulted from identifying that the PR abilities to think clearly and retain a sense of humour were the key aspects of the protective effect of PR upon sleep-related depression. These results have not been previously reported for patients with PCa and are congruent with the previous finding that deterioration of sleep quality is related to cognitive depression, or the inability to remember facts and make clear decisions, in patients with PCa [41].

Although some previous studies indicated that PR could exert protective effects against depression among patients with PCa [13], the current study extended that finding by identifying that the overall moderating effect of PR against sleep-related depression that was found at the total scale level was not uniform across all levels of deterioration in sleep quality; that is, when patients experienced a large decrease in their sleep quality, PR did not moderate their depression as much as it did when their deterioration in sleep quality was at an average or low level of severity. This interaction between the strength of patients’ PR and the severity of the deterioration they experienced in sleep quality resulted in specific PR moderating effects that were differentiated on the basis of deterioration in sleep quality. Thus, PR may be said to have a graduated moderating effect on sleep-induced depression in patients with PCa, a finding that extends previous reports of the overall protective effect of PR on these men’s depression.

### 4.1. Clinical Implications

The current study extended the current understanding of the treatment model for depression in patients with PCa by emphasising the need to consider sleep difficulties and PR in patients with PCa who are depressed. Firstly, sleep disturbances are among the key symptoms of major depressive disorder (MDD) [7], and their presence in patients with PCa may make those men even more vulnerable to other symptoms of MDD, as was found here. Therefore, formal clinical recognition and assessment of these men’s sleep difficulties should occur simultaneously with assessments of their depression and PR.

Secondly, various treatments have been suggested to assist patients with PCa to manage their stress, anxiety, and depression, including medication [42], psychotherapies such as mindfulness training [43], and exercise [44]. Alongside these, the training of patients in psychological resilience may also be valuable, particularly in regard to their sleep-related depression. PR training has been used to help people who face major stressful events to cope more effectively with those events [45], and it is likely that similar training can be applied to PCa patients whose quality of sleep has deteriorated since their diagnosis and treatment.

However, the key novel outcome of this study is that PR does not exert a uniform moderating effect upon depression arising from sleep difficulties but rather that there is an interaction between PR and the severity of the sleeping problems these men experience. This previously unreported interaction suggests that, while PR may be valuable for patients whose sleep problems are less than severe, other coping mechanisms may be needed for men who find that their sleep is much worse than it was prior to diagnosis and treatment. One such mechanism may be in the form of targeted medication, perhaps aimed at a short-term solution while PR training is being undertaken. Research evaluation of the effectiveness of this two-stage approach would provide valuable information for clinicians who work with patients with PCa.

### 4.2. Limitations

Limitations of this study include the geocultural aspects of the sample; no suggestion is made that these results will generalise to other populations and cultures. Similarly, the sample was self-selected, and no comments can be made regarding the generalisability of these findings to men who did not respond to the call for participants. The study used the retrospective pre-test, which is well-validated in the research literature but still remains a cross-sectional methodology. Extension of this study into a longitudinal model could help clarify if these findings are consistent among these men, or if particular periods after diagnosis are subject to greater sleep disturbance. Some strengths of this study include the use of well-validated scales that have been used in research with patients with PCa previously. Similarly, the application of moderation analysis enabled a greater degree of detail to merge regarding the effects of PR on depression arising from sleep difficulties.

## 5. Conclusions

In conclusion, the moderating effects of PR against depression among patients with PCa were confirmed and extended to those particular men who suffered from sleep difficulties after diagnosis and treatment. Moreover, these moderating effects were found to interact with the severity of sleep difficulties, which is a novel finding. The drilling down of analyses into the ‘core’ aspects of PR and sleep-related depression enabled much greater detail to be reported regarding these interactions than would have been possible using total scores from these instruments alone, showing that patients’ ability to think clearly when under pressure and see humour in adversity were paramount in helping them cope with sleep problems. This kind of study also provides the extra information needed to develop ‘individualised treatment’ models for depression in patients with PCa.

## Figures and Tables

**Figure 1 ijerph-19-08505-f001:**
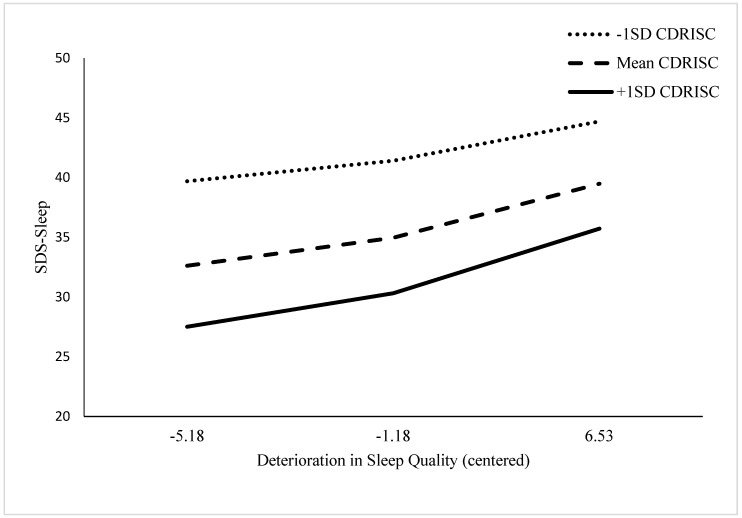
Interaction plot for SIS, CDRISC, and SDS-Sleep scores for 55 patients with PCa.

**Figure 2 ijerph-19-08505-f002:**
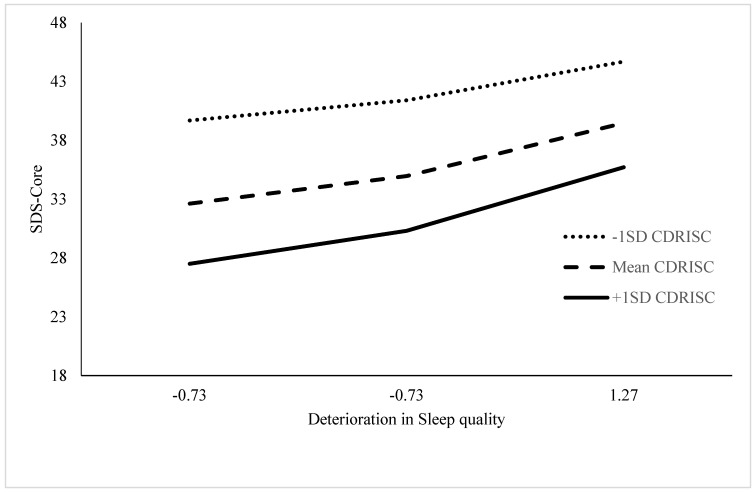
Interaction plot for decreases in sleep quality Item 1, CDRISC-Core, and SDS-Core scores for 55 patients with PCa.

**Table 1 ijerph-19-08505-t001:** Background and scale data for a sample of 55 patients with prostate cancer who reported a deterioration in sleeping quality following diagnosis and treatment.

Variable	Sample Characteristics
Age	*M* = 73.50 years (SD = 7.14 yr), range = 44 to 87 years
Relationship Status	
With wife/partner	72.9%
Widowed	6.8%
Divorced/separated	11.9%
Never married/partnered	8.5%
Time since diagnosis	*M* = 61.54 mo (SD = 24.27 mo), range = 1 to 197 mo
Treatments received	
Radiotherapy	27.8%
Surgery	7.4%
Hormone therapy	11.1%
Combinations	50.0%
Surveillance	3.1%
Current treatment	
Radiotherapy	52.6%
Hormone therapy	31.6%
Combinations	10.5%
Surveillance	5.3%
Present status	
Cancer still present, undergoing treatment	37.3%
In remission (no signs)	37.3%
Cancer recurring after previous treatment	25.4%
SDS	*M* = 36.14 (SD = 8.56), range = 20–57
SDS-Sleep	*M* = 31.72 (SD = 7.45), range = 18–50
SDS-Core	*M* = 11.02 (SD = 7.45), range = 18–50
CDRISC	*M* = 77.42 (SD = 14.51), range = 41–100
CDRISC-Core	*M* = 6.59 (SD = 1.64), range = 0–8
ISI change	*M* = 5.93 (SD = 5.40), range = 1–21
ISI Item 1 change	*M* = 0.73 (SD = 0.04), range = 1–4

SDS = Zung Self-Rating Depression Scale; SDS-Sleep = SDS minus two sleep-related items; SDS-Core = core items that contributed most to Cronbach alpha score; CDRISC = Connor–Davidson Resilience Scale; CDRISC-Core = items 6 and 141; ISI change = Insomnia Severity Index mean change from diagnosis/treatment to survey; ISI Item 1 change = SIS Item 1 mean change from diagnosis/treatment to survey.

**Table 2 ijerph-19-08505-t002:** Item–total correlations and Cronbach alpha for SDS-Sleep items from 55 patients with PCa (total scale alpha = 0.818).

SDS-Sleep Items	Corrected Item–Total Correlation	Alpha If Item Deleted
1. I feel downhearted and blue.	0.469	0.807
2. Morning is when I feel the best.	0.194	0.827
3. I have crying spells or feel like it.	0.264	0.817
4. I eat as much as I used to.	0.422	0.809
5. I still enjoy sex.	0.190	0.825
6. I notice that I am losing weight.	0.180	0.820
7. I have trouble with constipation.	0.332	0.813
8. My heart beats faster than usual.	0.172	0.819
**9. My mind is as clear as it used to be.**	**0.585**	**0.797**
**10. I find it easy to do the things I used to.**	**0.610**	**0.796**
11. I am restless and can’t keep still.	0.389	0.810
12. I feel hopeful about the future.	0.480	0.805
13. I am more irritable than usual.	0.417	0.810
**14. I find it easy to make decisions.**	**0.614**	**0.795**
**15. I feel that I am useful and needed.**	**0.570**	**0.798**
**16. My life is pretty full.**	**0.515**	**0.803**
17. I feel that others would be better off if I were dead.	0.235	0.818
**18. I still enjoy doing the things I used to.**	**0.692**	**0.794**

SDS-Sleep = SDS minus two sleep-related items.

## Data Availability

The anonymised data presented in this study are available on request from the corresponding author. The data are not publicly available due to patient confidentiality.

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
