# Peer review of "Which Aspects of Psychological Resilience Moderate the Association between Deterioration in Sleep and Depression in Patients with Prostate Cancer?"

_ijerph, 2022, doi:10.3390/ijerph19148505_

Round 1

Reviewer 1 Report

The paper is well written but I suggest an effort to link the content as suggested in the reviewer comments.

Reviewer comments

General comments:

The manuscript article addresses an important link between sleep patterns and depression among patients diagnosed with ca Prostrate. The manuscript is well written and the flow is easy to follow. My comments are the following:

1.     The title is in a form of a question, and the results should answer that question by clearly identifying the aspects of psychological resilience, and this must stand out in both the abstract and the conclusion of the paper.

2.     It would make it easier if the aspects of psychological resilience were stated, and the one/s which moderated the association would be singled out

3.     The authors state that ‘Psychological Resilience moderated the depressive effect of sleep deterioration following diagnosis/treatment’, which is confusing because when the participants were recruited they had already been diagnosed and were receiving treatment.

4.     Deterioration of sleep suggests that there was a baseline of some sort, from which the deterioration/getting worse was measured. The authors must re-think this and indicate if they were measuring sleep deterioration or poor sleeping patterns/sleeplessness

Author Response

Reviewer comments

General comments:

The manuscript article addresses an important link between sleep patterns and depression among patients diagnosed with ca Prostrate. The manuscript is well written and the flow is easy to follow. My comments are the following:

  1. The title is in a form of a question, and the results should answer that question by clearly identifying the aspects of psychological resilience, and this must stand out in both the abstract and the conclusion of the paper.

Author response: We have added the details about which aspects of psychological resilience were the key factors, and this information now appears in the Abstract and the Conclusion.

  1. It would make it easier if the aspects of psychological resilience were stated, and the one/s which moderated the association would be singled out.

Author response: we have now defined this aspect on lines 243 and 244 so that the specific aspects of resilience are clear to the reader.

  1. The authors state that ‘Psychological Resilience moderated the depressive effect of sleep deterioration following diagnosis/treatment’, which is confusing because when the participants were recruited they had already been diagnosed and were receiving treatment.

Author response: Thank you. We have revised that section now (lines 27-31) to make our meaning clearer.

  1. Deterioration of sleep suggests that there was a baseline of some sort, from which the deterioration/getting worse was measured. The authors must re-think this and indicate if they were measuring sleep deterioration or poor sleeping patterns/sleeplessness.

Author response: Agreed. We have revised line 34 to include the words “sleep deterioration that was reported by prostate cancer patients”.

Reviewer 2 Report

This is an interesting study that explores the moderating effect of psychological resilience on the depressive effects of sleep difficulties in prostate cancer patients after diagnosis and treatment of the disease. The results obtained are clinically relevant for the holistic therapeutic approach of these patients.

However, the article needs some revisions and clarifications before being considered for publication:

Methods

The target group consisted of 55 patients diagnosed with prostate cancer. However, the group is very heterogeneous in terms of age (range 44-87 years old), disease duration (range 1-197 months), disease status (disease present, in remission, recurrence) and treatment applied. The authors need to clarify whether this heterogeneity could influence the scores obtained in the questionnaires applied and how this influence was taken into account.

For example, the ISI Questionnaire was completed by participants at two time points: “How they felt during the last two weeks” and “How they felt before getting their diagnosis”. To what extent can the answers regarding the second moment investigated be considered reliable given that the study group also included patients diagnosed 197 months ago?

Bibliographic reference 37 does not seem to be correctly placed since it refers to a study of women suffering from breast cancer (p. 4).

Limitations

At the end of this section it is mentioned that: “The scales used are well-validated, and have been used in research with PCa patients previously. Similarly, the application of moderation analysis enabled a greater degree of detail to merge regarding the effects of PR upon depression arising from sleep difficulties”. The authors need to clarify why these issues are considered limitations of their study.

Discussion

This section should be developed in such a way that the results obtained in this study are discussed in the context of data from the literature. 

Conclusions

In this section, reference to Table 2, which appears in the Results section as well as the bibliographic citation, should be avoided.

Author Response

Comments and Suggestions for Authors

This is an interesting study that explores the moderating effect of psychological resilience on the depressive effects of sleep difficulties in prostate cancer patients after diagnosis and treatment of the disease. The results obtained are clinically relevant for the holistic therapeutic approach of these patients.

However, the article needs some revisions and clarifications before being considered for publication:

Methods

The target group consisted of 55 patients diagnosed with prostate cancer. However, the group is very heterogeneous in terms of age (range 44-87 years old), disease duration (range 1-197 months), disease status (disease present, in remission, recurrence) and treatment applied. The authors need to clarify whether this heterogeneity could influence the scores obtained in the questionnaires applied and how this influence was taken into account.

For example, the ISI Questionnaire was completed by participants at two time points: “How they felt during the last two weeks” and “How they felt before getting their diagnosis”. To what extent can the answers regarding the second moment investigated be considered reliable given that the study group also included patients diagnosed 197 months ago?

Author response: This is a reasonable question, and one that we gave thought to when analyzing the data. However, there were no significant associations with the demographic variables and the dependent variables. To clarify that point, we have now reiterated the analyses that were undertaken and stated the p value obtained (lines 182-185).

Bibliographic reference 37 does not seem to be correctly placed since it refers to a study of women suffering from breast cancer (p. 4).

Author response: Apologies. We have now replaced that with the correct reference.

Limitations

At the end of this section it is mentioned that: “The scales used are well-validated, and have been used in research with PCa patients previously. Similarly, the application of moderation analysis enabled a greater degree of detail to merge regarding the effects of PR upon depression arising from sleep difficulties”. The authors need to clarify why these issues are considered limitations of their study.

Author response: Thanks—these were strengths of the methodology, but we neglected to state that. We have now revised that section (lines 333-334) to state that “Some strengths of this study include” etc.

Discussion

This section should be developed in such a way that the results obtained in this study are discussed in the context of data from the literature. 

Author response: we have revised lines 280-307, making comparisons between our findings and those from the previous literature, and indicating how the current study extended the previous literature.

Conclusions

In this section, reference to Table 2, which appears in the Results section as well as the bibliographic citation, should be avoided.

Author response:  We have now removed those two aspects of the Conclusion.

Reviewer 3 Report

This is an important and informative study that evaluated associations between sleep disturbance, depression, and psychological resilience among men with prostate cancer. While this study is unique, there have been other studies that have evaluated co-occurring sleep disturbance and depression in this patient population (e.g., https://www.ncbi.nlm.nih.gov/pmc/articles/PMC7970199/pdf/10.1177_15579883211001201.pdf). This paper would be greatly strengthened with the inclusion of such studies in comparison to the findings in this study in the Discussion section. It would also be preferred for the term “prostate cancer patients” to be rephrased to “patients with prostate cancer” to avoid labeling individuals as their cancer instead of as an individual first.

Again, overall, this was a well conducted study and well written manuscript with strong implications for clinical practice.

Author Response

Comments and Suggestions for Authors

This is an important and informative study that evaluated associations between sleep disturbance, depression, and psychological resilience among men with prostate cancer. While this study is unique, there have been other studies that have evaluated co-occurring sleep disturbance and depression in this patient population (e.g., https://www.ncbi.nlm.nih.gov/pmc/articles/PMC7970199/pdf/10.1177_15579883211001201.pdf). This paper would be greatly strengthened with the inclusion of such studies in comparison to the findings in this study in the Discussion section. It would also be preferred for the term “prostate cancer patients” to be rephrased to “patients with prostate cancer” to avoid labeling individuals as their cancer instead of as an individual first.

Author response: Thanks to the reviewer for this suggestion. We have now included information from the study recommended by the Reviewer in the Discussion section, describing how those findings relate to the current data.

We have changed “prostate cancer patients” to patients with prostate cancer” throughout the ms as requested.

Again, overall, this was a well conducted study and well written manuscript with strong implications for clinical practice.

Round 2

Reviewer 2 Report

I thank the authors for carefully reviewing their manuscript. In the current version, the manuscript is much improved compared to the previous version.

I suggest two more minor changes- please see the file attached.

Author Response

We have made the two changes to the ms following the Reviewer's helpful suggestions.